# Shapley Is Not All You Need: Sobol's Total Indices for Feature Selection and Performance Loss Estimation

## Abstract

The selection of pertinent features constitutes a pivotal step in developing interpretable machine learning models, particularly when handling high-dimensional data, where the combinatorial interactions among features must be considered. The Shapley value, a concept originating from cooperative game theory, has gained recognition as a method for quantifying feature importance. However, the Shapley value often **fails to precisely** reflect the variance reduction that occurs when a feature is removed from the model. As the number of features increases, these challenges are further exacerbated by the **high computational complexity** of computing the exact Shapley value. Additionally, the common approximation techniques used to calculate the Shapley value are **not model-agnostic**. To address these gaps, we propose utilizing Sobol's total indices, a variance-based sensitivity analysis technique, as a more efficient and robust alternative to Shapley values. In this paper, we present both theoretical and empirical studies comparing these two methods. Sobol's total indices provide several key advantages. It captures both main effects and interactions, offering a more **accurate** importance measure than Shapley values. Its computation scales **linearly** with the number of features, making it suitable for high-dimensional problems. Additionally, it is derived from the data itself, ensuring complete **model-agnosticism**. Experiments on synthetic and real-world datasets demonstrate that feature selection using Sobol's total indices achieves better predictive performance than Shapley-based selection while requiring significantly less computational time. Our findings suggest that Sobol's total indices are a promising alternative to Shapley values, offering greater computational efficiency, comprehensiveness in accounting for interactions, and robustness in estimating variance. This represents a favorable substitute, particularly for high-dimensional feature selection. Code for the empirical experiments is provided in supplementary materials.

## 1 Introduction

In the era of big data and high-dimensional datasets, developing interpretable machine learning models that can elucidate the relationship between input features and model predictions has become increasingly important Molnar (2020); Murdoch et al. (2019). Feature selection is a fundamental aspect of building effective machine-learning models. It involves identifying the most relevant features that contribute to the predictive power of a model, thereby enhancing its performance and interpretability while reducing complexity and overfitting Guyon & Elisseeff (2003); Chandrashekar & Sahin (2014). Among the various techniques proposed for feature selection, methods that quantify each feature's importance or contribution to the model's predictions have gained significant attention Ribeiro et al. (2016); Molnar et al. (2020). One such method that has gained widespread popularity in recent years is the Shapley value, derived from cooperative game theory Shapley et al. (1953). The Shapley value assigns a unique importance value to each feature by considering its marginal contribution to the model's predictions across all possible coalitions of features Lundberg & Lee (2017). Due to its robust theoretical foundation, this approach has been successfully applied to various machine learning models, including tree-based methods Lundberg et al. (2020), neural networks Shrikumar et al. (2017), and kernel methods Song et al. (2016). Shapley values belong to variance-based feature-selection methods, which are special in feature selection due to a few merits. **First**, they offer unparalleled levels of explainability, which is crucial for understanding and interpreting the contributions of individual features in a model. Traditional feature-selection methods primarily focus on improving model performance but often lack clear, interpretable insights into

| Criteria | Sobol's Total Indices | Shapley Values |
|----------|----------------------|----------------|
| Variance Capture | **Accurate** | **Over- or Under-estimate** (due to averaging) |
| Time Complexity | **Lower** | **Higher** (due to factorial growth) |
| Model Agnostic | **Yes** | **Approximation methods are model-dependent** |

**Table 1.** Comparison of Sobol's Total Indices and Shapley Values

why certain features are selected or discarded. Comparatively, variance-based methods decompose the variance of the model output attributed to each feature and their interactions, providing a clear understanding of how each feature influences the outcome and making it easier for stakeholders to understand and trust the feature selection results. **Second**, the variance-based methods are purely based on data and can be applied to any machine learning model, making them versatile tools for feature selection across different domains and applications. This model-agnostic nature is not always present in other feature-selection methods, which might be tailored to specific types of models (e.g., decision trees or linear models). **Third**, these methods are capable to capture and quantify interaction effects among features. Traditional feature selection methods often consider features in isolation or through simple pairwise interactions, potentially missing out on complex, higher-order interactions. On the other hand, the variance-based methods explicitly account for the contribution of interactions among features, providing a more comprehensive understanding of feature importance. **Last but not least**, variance-based methods are particularly effective at estimating the performance loss when a feature is selected to be excluded from the model, which is a critical aspect of feature selection. This ability to quantify the impact of excluding features helps in understanding the robustness and resilience of the model. Overall, variance-based methods are particularly useful in high-dimensional settings, where understanding the interplay between features is crucial for model interpretability and performance optimization.

However, despite its theoretical elegance and interpretability, the Shapley values suffer from several major limitations. *First*, inaccuracy arises because the Shapley values cannot correctly capture the lost variance when certain features are excluded. To satisfy the Efficiency Axiom Roth (1988), the Shapley value of a feature only partially reflects its interaction effect with other features. Consequently, when that feature is excluded from model training, the entire interaction effect and its first-order effect are lost, leading to inaccurate results when using the Shapley value to measure the variance lost due to feature exclusion. *Second*, the high complexity of Shapley values is a significant drawback. Its computational complexity scales exponentially with feature count Štrumbelj & Kononenko (2014), making Shapley values impractical for high-dimensional data analysis in fields like genomics Libbrecht & Noble (2015), finance Heaton et al. (2017), and computer vision

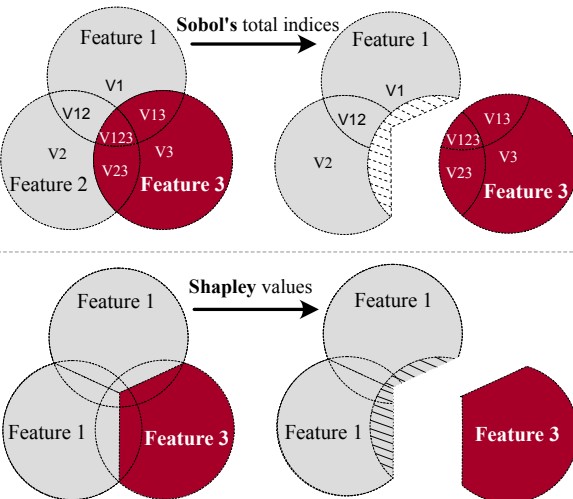

**Fig. 1.** Variance lost when excluding a feature. Sobol's total index captures all variance attributed to the excluded feature, while the Shapley value retains part of the interaction effect with remaining features, even though these interactions no longer exist.

Krizhevsky et al. (2012). *Third*, due to its high computational complexity, modern approaches often approximate the Shapley value, but these approximations are typically model-dependent Lundberg & Lee (2017); Lundberg et al. (2018); Sundararajan et al. (2017). Even so-called "model-agnostic" approaches still require model predictions despite not needing access to the model internals.

To address this limitation, we propose using Sobol's total index as an alternative approach for quantifying feature importance and performing feature selection. Sobol's indices are a variance-based sensitivity analysis technique originally developed in uncertainty quantification Sobol (2001). They decompose the variance of the model output into contributions from individual features and their

interactions, providing a comprehensive understanding of each feature's importance. Specifically, the total Sobol's index for a given feature quantifies its overall importance, encompassing both its main effect and its interactions with other features Saltelli (2002). By decomposing the total variance of the model output, Sobol's indices reflect the true importance of features within the predictive model. This capability is crucial for understanding the intricate interplay between features and their combined impact on the model's performance, making Sobol's total index particularly well-suited for feature selection and mitigating the inaccurate estimation of the lost variance when using the Shapley value Wei et al. (2015). The difference between how Sobol's total indices and Shapley values decompose the variance is demonstrated in Fig. 1. Moreover, the computation of Sobol's indices scales linearly with the number of features, making them significantly more efficient than Shapley values for high-dimensional problems Sudret (2008). Also, the computation of Sobol's indices involves solely the variance of the target variable given the feature. This process is totally free of limitations from prediction models. The ability to accurately capture feature interactions, coupled with the computational advantage and the true model-agnostic nature, positions Sobol's total index as a promising alternative to Shapley values for feature selection. Specifically, this paper presents a comprehensive study comparing the performance of Sobol's total indices and Shapley values for feature selection across a diverse range of synthetic and real-world datasets. Our primary contributions include:

- **We provide a detailed comparison between Sobol's total index and the Shapley value, highlighting their capability of capturing the variance lost due to feature exclusion**. Through theoretical analysis, we illustrate how Sobol's total index accurately captures the variance lost due to feature exclusion, including both main and interaction effects. We explain the potential inaccuracies of the Shapley value in estimating the lost variance due to its averaging process, which can lead to overestimation or underestimation.

- **The performance difference on feature selection tasks is evaluated through empirical experiments.** Real-world and synthetic datasets are employed to demonstrate that Sobol's total index achieves comparable or superior performance.

- **Time complexities of calculating or estimating the Shapley values and Sobol's total index are compared.** This paper includes a detailed empirical evaluation of the running time for calculating Sobol's total indices and the Shapley values across different datasets and model types.

## 2 PROBLEM FORMULATION

The central task in this study is feature exclusion - the identification of the least relevant feature to remove from the dataset while minimizing the impact on model performance. Specifically, the goal is to quantify the performance loss after excluding a feature and to determine a method that best approximates this loss.

Given a trained model, removing any feature might affect its prediction performance. Our objective is to find the method (between Shapley values and Sobol's total indices) that yields the most accurate approximation of the true performance loss caused by excluding a feature. More formally, for a model $Y = f(X)$, where $Y$ is the output variable and $X = \{x_1, x_2, \ldots, x_n\}$ is the set of input features, the task is to exclude a feature $x_i$ and quantify the resulting change in model performance

$$\Delta(x_i) = Eval(f(X)) - Eval(f(X_{\sim i})). \tag{1}$$

Here, $Eval()$ stands for the model performance, such as accuracy for classification models and $R^2$ for regression models. $X_{\sim i}$ means all the elements in set $X$ except for the $i$-th element.

We aim to minimize the discrepancy between the approximated loss (as predicted by Shapley values or Sobol's total indices) and the real influence observed after removing the feature from the model. Specifically, let the true performance change after removing $x_i$ be $\Delta(x_i)$, and the approximation by a given method be $\hat{\Delta(x_i)}$. We seek to minimize the difference $|\Delta(x_i) - \hat{\Delta(x_i)}|$ across all features, ensuring that the method selected provides the most accurate measure of feature importance.

This formulation leads us to a natural comparison between Shapley values and Sobol's total indices to determine which method best captures the true model impact of excluding a feature and supports better feature selection decisions.

**Sobol's Indices** are metrics for global sensitivity analysis that quantify each input variable's contribution to a model's output variability. They are essential for assessing how inputs affect a model's output variance Soboĺ (1993). This method is useful for complex models with non-linear input-output relationships. Sobol's indices decompose the output variance into contributions from each input and their interactions, capturing both first-order (individual) and higher-order (interaction) effects Saltelli (2002). For $Y = f(X_1, X_2, ..., X_d)$, the total variance $\mathbb{V}(Y)$ is decomposed as:

$$\mathbb{V}(Y) = \Sigma_{i=1}^d V_i + \Sigma_{1 \leq i < j \leq d} V_{ij} + \cdots + V_{12...d} \tag{2}$$

The *first-order Sobol's index* is $S_i = \frac{V_i}{\mathbb{V}(Y)}$, while the *total Sobol's index* $S_{T_i}$ includes all variance contributions involving $X_i$: $S_{T_i} = \frac{V_i + \Sigma_{j \neq i} V_{ij} + \cdots + V_{12...d}}{\mathbb{V}(Y)}$. For machine learning, input-output data can be used directly. Sobol's total index is:

$$S_{T_i} = 1 - \frac{\mathbb{V}(\mathbb{E}(Y|X_{\sim i}))}{\mathbb{V}(Y)} = \frac{\mathbb{E}_{\mathbf{X}_{\sim i}} (\mathbb{V}_{X_i} (Y \mid \mathbf{X}_{\sim i}))}{\mathbb{V}(Y)}. \tag{3}$$

**Shapley Value** Shapley et al. (1953) is used to fairly distribute the total gain generated by a coalition of players based on their contributions. This method ensures that both main effects and interaction effects are accounted for, providing a holistic measure of feature importance Lundberg & Lee (2017). Given a model $Y = f(X_1, X_2, ..., X_d)$, the Shapley value for feature $X_i$ is defined as:

$$\phi_i = \sum_{S \subseteq N \backslash \{i\}} \frac{|S|!(|N| - |S| - 1)!}{|N|!} [f(S \cup \{i\}) - f(S)] \tag{4}$$

where $N$ is the set of all features, $S$ is a subset of $N$ not containing $X_i$, and $f(S)$ denotes the model output when only the features in $S$ are used. The Shapley value computes the average marginal contribution of a feature across all possible subsets, ensuring a fair distribution of importance scores. To calculate the exact Shapley values, we need to iterate through all subsets that include the target feature and calculate the variance accounted for the interaction effect involving all elements in each subset. This variance can be computed by Sobol's higher-order effect. The variance brought by the interactions between two variables can be computed by removing the first-order effects of the two variables from the first-order effect of the two-variable subset. That is to say $V_{i \times j} = V_{ij} - V_i - V_j$, where $V_{i \times j}$ is the second-order interaction effect of variables $X_i$ and $X_j$. Similarly, the variance brought by the third-order interaction of three variables $X_i, X_j$, and $X_k$ can be written as:

$$V_{i \times j \times k} = V_{ijk} - V_{ij} - V_{ik} - V_{jk} - V_i - V_j - V_k \tag{5}$$

From Eq. 2, we can see that the total variance explained by the input variables is decomposed to the first-order effects of the variables and the interaction effects of all orders. The exact Shapley value of one feature can be calculated by:

$$\phi_i = V_i + \frac{\Sigma_{j \neq i} V_{i \times j}}{2} + \frac{\Sigma_{i \neq j \neq k} V_{i \times j \times k}}{3} + \cdots + \frac{V_{1 \times 2 \times ... \times d}}{d} \tag{6}$$

Calculating exact Shapley values is computationally infeasible for large feature sets due to the factorial growth of subsets. To address this challenge, several approximation methods have been developed. These include Monte Carlo Sampling, which approximates Shapley values by randomly sampling subsets of possible coalitions and then estimating the Shapley value from these samples Castro et al. (2009). Kernel SHAP uses a weighted linear regression approach to approximate Shapley values, particularly efficient for linear models Lundberg & Lee (2017). Tree SHAP is an algorithm specifically designed for tree-based models, leveraging the structure of trees to reduce the complexity of Shapley value computation Lundberg et al. (2020). These techniques significantly reduce the computational burden, making Shapley values feasible for large datasets. However, these methods are often model-dependent, requiring either access to the model's inner workings or its predictions to speed up the computation.

The actual performance loss for regression models and classification models are scaled differently. On the one hand, $R^2$ measures how well a regression model's predictions approximate the actual data points, where 1 means perfect predictions and 0 means that the model performs as well as simply predicting the mean of the target variable. On the other hand, accuracy for a classification model also ranges from 0 to 1, where 1 indicates perfect predictions. A random guessing classification

model has an expected accuracy of $\frac{1}{c}$ instead of 0, where $c$ is the number of classes. For example, random guessing would typically yield $0.5$ accuracy for a binary classification task. Noticing that $R^2 = 0$ and $accuracy = 0$ have different meanings, we standardize the accuracy of a classification model such that a model performing as well as a random guessing model has an accuracy of 0.

$$\bar{\text{Accuracy}} = \frac{\text{Accuracy} - \frac{1}{c}}{1 - \frac{1}{c}} = \frac{c * \text{Accuracy} - 1}{c - 1} \tag{7}$$

$$\bar{\Delta}(x_i) = \begin{cases} R^2 & \text{for regression tasks} \\ \bar{\text{Accuracy}} & \text{for classification tasks} \end{cases} \tag{8}$$

Given Equations 3, 6, and 8, our objective is to prove that for all features in the input sets:

$$|\bar{\Delta}(x_i) - \phi_i| \leq |\bar{\Delta}(x_i) - S_{T_i}| \tag{9}$$

## 3 THE ADVANTAGE OF SOBOL'S TOTAL INDICES

Past research indicates that the Shapley values are preferable for feature selection, as they consider all possible combinations of features, thereby providing a comprehensive insight into feature contributions Lundberg & Lee (2017). Conversely, other studies argue against the use of Sobol's total index, citing two primary reasons: (1) it fails to satisfy the efficiency axiom (the additive assumption), and (2) when features are positively correlated, the sum of Sobol's total indices is less than the total variance Song et al. (2016). In this section, we will demonstrate that Sobol's total index is superior to the Shapley value due to its ability to quantify lost variance more accurately and lower computational cost.

**More accurate estimation of the variance loss.** The most critical advantage of Sobol total indices is their ability to accurately capture the variance loss when a feature is excluded from the model. When a feature is removed, all the variance explained by that feature and its interactions with other features is lost. Sobol total indices are designed to capture this total variance, including both main effects and interaction effects, providing a comprehensive measure of feature importance. In contrast, while theoretically rigorous in distributing contributions among features, the Shapley value can overestimate or underestimate the lost variance due to its averaging process across all subsets Owen & Prieur (2017). The Shapley value calculates the marginal contribution of each feature by averaging its impact across all possible subsets, which can lead to inaccuracies in capturing the true variance loss when features are excluded. This averaging process may not fully account for complex interactions between features, leading to potential biases in the importance measures.

By focusing on the total variance, Sobol indices provide a more precise and reliable assessment of feature importance, particularly in models where interactions play a significant role. This accurate capture of lost variance is crucial for developing robust and interpretable machine learning models, ensuring that important features are correctly identified and leveraged. From Figure 1, we can intuitively observe why Sobol's total index evaluates the lost variance due to excluding a feature more accurately than the Shapley value. In the Venn diagram with three overlapping circles representing three features, the areas of overlap indicate the interactions between features. Sobol's total indices capture each feature's individual contributions and interactions. For instance, the total index

| A | B | A XOR B |
|---|---|---------|
| 0 | 0 | 0 |
| 0 | 1 | 1 |
| 1 | 0 | 1 |
| 1 | 1 | 0 |

**Table 2.** XOR function: Neither of the features has a first-order effect on the output, while the interaction can accurately predict the output.

for Feature 3 includes the variance explained by Feature 3 alone, the variance explained by the second-order interactions between Feature 3 and other features, and the variance explained by the third-order interaction of all three features. Thus, we have $Sr_{T_3} = V_3 + V_{1\times3} + V_{2\times3} + V_{1\times2\times3}$. When Feature 3 is excluded from the model due to feature selection, the variance explained by the rest of the model is

$$\mathbb{V}(Y|(x1, x2)) = \mathbb{V}(Y|(x1, x2, x3)) - S_{T_3} = V_1 + V_2 + V_{1\times2} \tag{10}$$

Contrarily, the Shapley value allocates the importance of each feature by averaging their contributions over all possible subsets of features. Therefore, $\phi_3 = V_3 + \frac{V_{1\times3}}{2} + \frac{V_{2\times3}}{2} + \frac{V_{1\times2\times3}}{3}$. The Shapley

value approach predicts that the variance explained by the rest of the model is:

$$\mathbb{V}(Y|(x1,x2)) = \mathbb{V}(Y|(x1,x2,x3)) - \phi_3 = V_1 + V_2 + V_{1\times2} + \frac{V_{1\times3}}{2} + \frac{V_{2\times3}}{2} + \frac{2 \cdot V_{1\times2\times3}}{3} \quad (11)$$

From Eq. 11, we can tell that the variance explained by the new model predicted by the Shapley value feature-selection approach accounts for the interactions that are not in the model anymore. Depending on the sign of the last three terms, this approach would overestimate or underestimate the new model's performance. We can bring this estimation inaccuracy of the Shapley-value-based feature selection approach to an extreme when the remaining feature set has no first-order effect on the variance of the output. Consider a function $Y = f(X_1, X_2)$. When $X_1$ and $X_2$ have no first-order effect, excluding either will result in the new model losing all its explaining power. A typical example of this function is the XOR function (Table. 2). Neither of the features is correlated with function output, but the interaction of the two features can fully predict the output. Sobol's total indices of both features are 1, suggesting that excluding either of the features would cause the full predicting power of the model. However, the Shapley values of the features suggest that the new model can still explain half of the variance in the output when excluding either one of the features.

**Refutation of Contemporary Criticisms on Sobol's Total Indices.** We mentioned above that Sobol's total indices were considered unsuitable for feature selection because they do not satisfy the efficiency axiom Owen (2014). We will discuss why the efficiency axiom is unnecessary and demonstrate that free of this nature helps accurately estimate the lost variance.

The efficiency axiom Roth (1988), a fundamental principle in cooperative game theory, asserts that the total value generated by a coalition of players should be fully distributed among them, such that $\Sigma_{i=1}^{n} \pi_i = 1$ where $\pi_i$ is the contribution of the $i$-th player. While this axiom is crucial for fair distribution in resource allocation problems, its application to feature selection in machine learning is both unnecessary and potentially harmful, and here is why. The efficiency axiom ensures that the sum of contributions of all players equals the total value of the coalition, making it highly relevant in scenarios where resources or rewards need to be distributed among participants. In feature selection, however, the goal is different. Rather than distributing resources within a system, we are concerned with evaluating the impact of excluding individual features from the model. The efficiency axiom does not naturally apply in this context because removing a feature from a model is not analogous to distributing resources among the remaining features. Instead, it focuses on understanding features' individual and collective contributions to the model's performance. For instance, when we exclude a feature in a machine learning model, we are interested in the change in the model's predictive power. This is not about redistributing the model's accuracy or variance among the remaining features but about assessing the importance of the removed feature. Thus, adhering to the efficiency axiom can distort this evaluation by imposing a constraint that is irrelevant to the actual task. As demonstrated by the XOR function example, if we allocate the features' contributions to the model's predictive power, both Sobol's total indices and the Shapley values indicate that the two features have the same contribution. $\phi_1 = \phi_2 = 0.5$, and $S_{T_1} = S_{T_2} = 1$. However, Sobol's total indices correctly suggest that excluding either feature will result in the loss of all predictive power, while the Shapley values fail to do so due to the limitation of the efficiency axiom.

Song et al. Song et al. (2016) also proved that there exists a joint distribution of features $X$ and function $f$ such that $\Sigma_{i=1}^{d} V_i > \mathbb{V}(Y) > \Sigma_{i=1}^{d} S_{T_i}$. This theorem has been traditionally considered as a reason why Sobol's total indices are not a good basis for feature selection. This phenomenon happens when the features are highly correlated with each other. However, this is actually how Sobol's total indices inherently recognize and handle redundancy. Consider a set of positively correlated features. If a feature is highly correlated with others, it provides less unique information. In this case, Sobol's total index will be lower for this feature, indicating that its exclusion will result in less variance loss. This aligns with practical expectations: a largely redundant feature should not be deemed critical, and its exclusion should not significantly impact the model's performance. This is also the rationale of feature selection based on feature correlations. In contrast, enforcing the efficiency axiom through the Shapley values would distribute the total variance among features without accounting for redundancy. This distribution can overstate the importance of highly-correlated features, leading to suboptimal feature selection results. By not imposing the efficiency axiom, Sobol's total indices offer a more realistic measure of feature importance, recognizing the diminishing returns of redundant information.

**Lower Time Complexity and Computational Efficiency.** One of the primary advantages of Sobol's total index over Shapley value is its lower time complexity and reduced computational expense. The computational cost of calculating Shapley values increases factorially with the number of features, making it impractical for high-dimensional datasets. Specifically, the exact computation of the Shapley value for a model with $d$ features requires evaluating $2^d$ possible subsets of features, leading to a time complexity of $O(2^d)$ Lundberg & Lee (2017). This exponential growth makes Shapley values computationally infeasible without resorting to approximations. In contrast, Sobol's Total Indices can maintain this nature with a linear time complexity. Given a dataset consisting of input features $X$ and the output $Y$ with $N$ data points, the time complexity of the calculation is on the order of $O(N \cdot d)$ Saltelli et al. (2010). This efficiency makes Sobol's total indices particularly suitable for high-dimensional problems and complex models with limited computational resources, making it a versatile tool for feature selection across a wide range of applications.

## 4 EMPIRICAL EXPERIMENTS

We implemented Sobol's total indices and the exact Shapley values algorithms in Python using standard libraries such as NumPy and Pandas. The experiments are carried out on a server with AMD Ryzen Threadripper PRO 5955WX 16-core CPU with 128GB RAM. **Datasets.** We utilize two synthetic datasets and four realistic datasets from UCI Lichman et al. (2013) to compare Sobol's total indices and the Shapley values for feature selection. They include: **1, Synthetic Correlated Dataset**: Four features and one output, all with a correlation of 0.9, designed to test the handling of highly correlated features. **2, Synthetic XOR Dataset**: Two binary features and one binary outcome representing the XOR function are used to assess the handling of interactions. **3, Diabetes**: Medical diagnostic measurements with a binary outcome indicating diabetes presence. **4, Wine**: Thirteen chemical properties of wines from three cultivars are used for multi-class classification. **5, Auto-MPG**: Automobile attributes predicting miles per gallon (MPG), used for regression. And **6, Concrete Compressive Strength**: Ingredients of concrete predicting compressive strength. These datasets provide a comprehensive evaluation across different types of data and tasks.

| Feature | Standardized Actual Performance Loss | Sobol's Total | Shapley |
|---|---|---|---|
| High-Correlation Feature 1 | 0 | 0 | 0.25 |
| High-Correlation Feature 2 | 0 | 0 | 0.25 |
| High-Correlation Feature 3 | 0 | 0 | 0.25 |
| High-Correlation Feature 4 | 0 | 0 | 0.25 |
| XOR Feature 1 | 1 | 1 | 0.5 |
| XOR Feature 2 | 1 | 1 | 0.5 |

**Table 3.** Comparison of Sobol's Total Indices and Shapley Values for Feature Importance. Top: Highly Correlated Dataset. Bottom: XOR Dataset

**Synthetic Datasets Analysis.**

We first utilize the Synthetic Correlated Dataset to demonstrate the capability difference between Sobol's total indices and the Shapley values on handling highly correlated features. From the top part of Table 3 we can observe that Sobol's total indices for all four features are all 0, while their Shapley values are all 0.25. The actual performance loss of the regression model is 0 when excluding one of the four input features. This indicates that Sobol's total indices accurately identify redundant features, indicating that any of these features

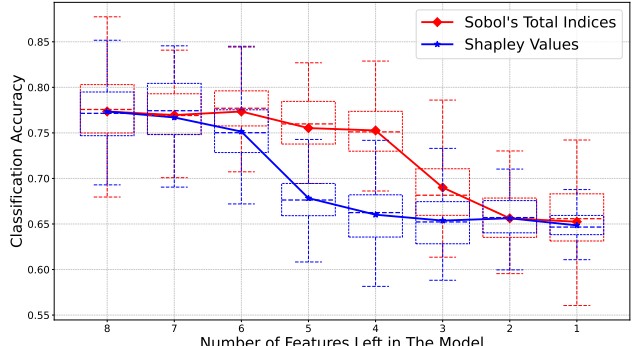

**Fig. 2.** Accuracy Change of Logistic Regression on the Diabetes Dataset As the Feature Number Decreases.

could be removed without loss of variance, aligning with the expected behavior in cases of high feature redundancy. Comparatively, the Shapley values, enforced by the efficiency axiom, overestimate the individual contributions of highly correlated features, exaggerating their importance in feature selection tasks. Then, we calculate the Sobol's total indices and the Shapley values of the two features in the Synthetic XOR Dataset. Sobol's total indices for the two features are both 1, indicating that excluding any of them would make the model lose all its predictive power. The Shapley values, which average the variance accounted for the interaction effect, are both $0.5$. This suggests that with only one feature, the model can still account for half of the variance in the XOR function. Obviously, this conclusion is wrong since either feature is independent of the XOR function output.

**Realistic Datasets Analysis.**

To demonstrate the superiority of Sobol's total indices over the Shapley values, we test the feature-selection performance of both algorithms with four realistic datasets, two for classification and two for regression. We utilize Random Forest Classifiers, Decision Tree Classifiers, and Logistic Regression Classifiers for classification tasks, and use Random Forest Regressors, Decision Tree Regressors, and Linear Regression for regression tasks. Fig. 2 illustrates the accuracy change of a Logistic Regression model over the Diabetes Dataset. As the number of features decreases from 8 to 1, Sobol's total indices and the Shapley values suggest two different exclusion orders in Table. 4. We can observe that the model suffers from less variance loss when following Sobol's order. The sudden drop in prediction accuracy happens when the number of features drops to 3. When excluding features based on Shapley's order, the sudden drop appears as early as when the feature number drops to 5. This is because a relatively important feature is excluded too early due to underestimating its importance or overestimating the other features' importance.

Sobol's total indices are particularly well-suited for regression tasks because they directly evaluate the variance explained by the model due to individual features and combinations of features. This property makes Sobol's total indices highly relevant for understanding the relative importance of features in explaining the variance of the target variable in regression models. Fig. 3 shows the $R^2$ score change of a Random Forest Regressor over the Auto-MPG Dataset.

| Sobol | Shapley |
|---|---|
| DiabetesPedigreeFunction | Age |
| Insulin | BMI |
| Glucose | SkinThickness |
| BMI | BloodPressure |
| SkinThickness | Glucose |
| BloodPressure | Pregnancies |
| Pregnancies | Insulin |
| Age | DiabetesPedigreeFunction |

**Table 4.** Orders of Feature Exclusion

We can observe that Sobol's total indices can maintain a relatively high performance until the number of features drops under 3. This is due to the high interaction effects existing among the three features left. The Shapley values experience that performance drop when the feature number changes from 6 to 5 because one of the features with a significant high-order interaction effect is excluded there. This incorrect feature-selection decision comes from the underestimation of the importance of the highly interacted features, which is due to the averaging process of the Shapley values. The two algorithms' performance with multiple datasets and machine-learning models show similar results, as demonstrated in Tables 5, 6, and 7.

**Runtime Analysis.** To validate whether Sobol's total indices are more scalable than the Shapley value, we measure the runtime of calculating both given various instance numbers and feature numbers. The results are illustrated in Figure 4. When the number of instances is fixed, the runtime of Sobol's total indices remains consistently low across different numbers of features, reflecting its linear time complexity. Contrarily, the runtime of the Shapley value increases exponentially as the number of

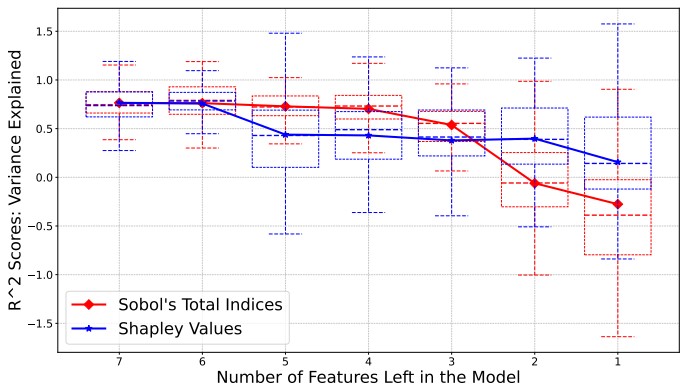

**Fig. 3.** $R^2$ Change of Random Forest Regressor on Auto-MPG.

features increases, illustrating
its combinatorial time complexity. For example, with 8 features and 100 instances, Sobol's total indices required only 0.085 seconds, while the Shapley Values took 9111.088 seconds. This stark contrast underscores the substantial difference in computational efficiency and scalability between the two methods. We also investigated the runtime performance of both algorithms with a fixed number of features while varying the number of instances to understand how each algorithm scales with increasing data volume. As shown in the right sub-figure of Fig. 4, both Sobol's total indices and the Shapley values exhibited an increase in runtime as the number of instances increased, which is expected due to the additional computational overhead associated with processing more data. However, the rate of increase was remarkably different between the two algorithms. Sobol's total indices demonstrated a much smaller slope and consistently outperformed the Shapley values in terms of runtime across all instances. The results indicate that Sobol's total indices offer superior runtime performance compared to Shapley Values for feature selection tasks. This advantage becomes more pronounced with larger datasets, where Shapley Values exhibit significant computational overhead. Therefore, when considering computational efficiency, Sobol's total indices emerge as the preferred choice for feature selection tasks, especially with high-dimensional datasets.

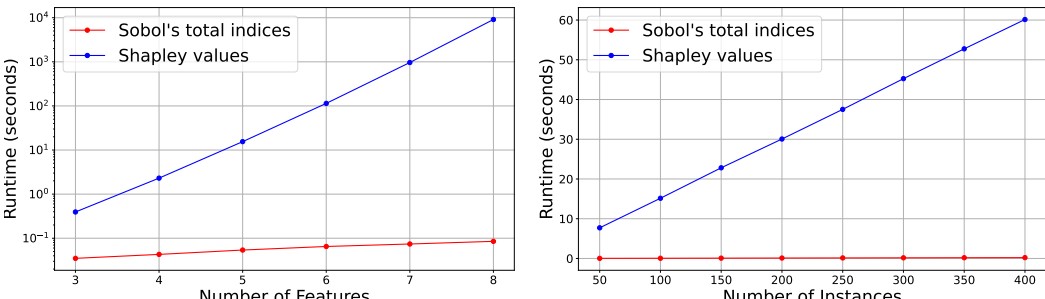

**Fig. 4.** Runtime Comparison of Sobol's Total Indices and Shapley Values for Different Instance Numbers and Feature Numbers. Left: Instance Number Fixed to 100 While Feature Number Changes. Right: Feature Number Fixed to 5 While Instance Number Changes.

## 5 RELATED WORK

**Feature selection** is a critical step in the machine learning pipeline, aimed at improving model performance by identifying the most relevant features while reducing the dimensionality of the dataset. This process not only enhances computational efficiency but also aids in the interpretability of the model. There are various methodologies and frameworks for feature selection, which can be broadly classified into three categories: **(1) Filter methods** apply statistical techniques to evaluate the relevance of each feature independently of the learning algorithm. These methods are generally computationally efficient and scalable to high-dimensional datasets. Common filter techniques include: correlation coefficent Guyon & Elisseeff (2003), mutual information Battiti (1994), Chi-Square test Liu & Setiono (1995), and variance thresholding Roffo et al. (2015). These methods typically evaluate features in isolation without considering higher-order interactions. They also do not provide a clear estimate of how the exclusion of features impacts model performance. **(2) Wrapper methods** evaluate feature subsets based on the performance of a specific learning algorithm. These methods typically involve iterative search procedures to find the optimal feature subset, which makes them computationally intensive but often more accurate than filter methods. Some popular wrapper techniques include: recursive feature elimination Guyon et al. (2002), genetic algorithms Holland (1992), and forwards and backward selection Draper & Smith (1998). These methods do not explicitly quantify the interaction effects among the features and fail to provide a clear understanding of why features are selected or discarded. Also, they are highly dependent on the choice of the models and may be computationally expensive when the model-fitting is complex. **(3) Embedded methods** integrate the selection process within the construction of the machine learning model itself. These methods leverage the model's predictive capabilities to evaluate and select the most relevant features, typically during the training phase. Classic embedded methods include regularization techniques,

such as LASSO Tibshirani (1996) and Ridge Regression Hoerl & Kennard (1970), and decision tree-based methods, such as random forests Breiman (2001) and Gradient Boosting Machines Friedman (2001). These methods are tied to specific models and cannot be easily generalized across different model types. Also, regularization methods do not explicitly account for interactions between features. Tree-based methods can capture some interactions but do not quantify them separately.

## 6 Conclusion and Limitations

In this paper, we proposed using Sobol's total indices instead of Shapley Values for feature selection tasks. They have more accurate variance loss estimation, lower time complexity, and are model-agnostic. Our experiments demonstrated that they offer superior computational efficiency and better feature-selection performance compared to Shapley Values. Despite the promising results obtained in our experiments, several limitations should be considered: (**1**). Due to resource constraints and the computational complexity of the Shapley values, we could not conduct experiments on large datasets. As a result, the performance of Sobol's total indices and Shapley Values on high-dimensional datasets is not empirically tested. (**2**). While Sobol's total indices work better for feature selection tasks, they might not interpret a machine-learning model as well as the Shapley values do. Therefore, based on our findings, we conclude that Sobol's total indices are better suited for feature selection in machine learning applications. Future research could be conducted to explore ways to enhance the interpretability of Sobol's total indices.

| Number of Features | | 13 | 12 | 11 | 10 | 9 | 8 | 7 |
|---|---|---|---|---|---|---|---|---|
| | LR | 95.5 | 95.0 | 94.4 | 95.0 | 95.0 | 94.4 | 93.3 |
| Shapley | DT | 84.8 | 89.4 | 89.9 | 89.4 | 89.4 | 90.0 | 88.8 |
| | RF | 98.3 | 96.7 | 98.3 | 97.8 | 97.2 | 97.2 | 97.2 |
| | LR | 95.5 | 95.6 | 97.2 | 95.6 | 95.6 | 95.0 | 94.4 |
| Sobol | DT | 84.8 | 91.1 | 90.0 | 93.3 | 92.7 | 92.2 | 94.9 |
| | RF | 98.3 | 97.2 | 97.8 | 97.8 | 97.8 | 98.3 | 97.2 |

**Table 5.** The accuracy (%) comparison between the two methods on the Wine dataset, with the feature numbers shrinking from 13 to 7.
LR: Logistic Regression. DT: Decision Tree Classifier. RF: Random Forest Classifier

| Number of Features | | 8 | 7 | 6 | 5 | 4 | 3 | 2 |
|---|---|---|---|---|---|---|---|---|
| | LR | 0.278 | 0.283 | 0.289 | 0.300 | -0.019 | 0.015 | -0.104 |
| Shapley | DT | 0.455 | 0.438 | 0.459 | 0.545 | 0.503 | 0.424 | 0.121 |
| | RF | 0.740 | 0.728 | 0.748 | 0.741 | 0.728 | 0.606 | 0.322 |
| | LR | 0.278 | 0.285 | 0.294 | 0.300 | 0.201 | 0.113 | -0.068 |
| Sobol | DT | 0.455 | 0.597 | 0.570 | 0.562 | 0.527 | 0.454 | 0.119 |
| | RF | 0.740 | 0.749 | 0.748 | 0.742 | 0.728 | 0.606 | 0.324 |

**Table 6.** The $R^2$ score comparison between the two methods on the Concrete Compressive Strength dataset, with the feature numbers shrinking from 8 to 2.
LR: Linear Regression. DT: Decision Tree Regressor. RF: Random Forest Regressor

| Number of Features | | 14 | 13 | 12 | 11 | 10 | 9 | 8 | 7 |
|---|---|---|---|---|---|---|---|---|---|
| | LR | 79.8 | 77.6 | 78.9 | 75.4 | 70.1 | 64.8 | 60.6 | 56.4 |
| Shapley | DT | 76.2 | 75.4 | 74.8 | 73.2 | 66.9 | 64.3 | 60.6 | 55.1 |
| | RF | 85.2 | 83.7 | 85.1 | 79.6 | 73.2 | 65.4 | 62.5 | 58.7 |
| | LR | 79.8 | 80.9 | 79.4 | 77.2 | 70.2 | 64.8 | 58.8 | 55.1 |
| Sobol | DT | 76.2 | 78.0 | 76.8 | 75.4 | 66.9 | 64.5 | 58.8 | 53.4 |
| | RF | 85.2 | 86.4 | 85.2 | 81.3 | 76.7 | 64.7 | 60.6 | 57.2 |

**Table 7.** The accuracy (%) comparison between the two methods on the Adult dataset, with the feature numbers shrinking from 14 to 7.
LR: Logistic Regression. DT: Decision Tree Classifier. RF: Random Forest Classifier

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
