# OpenReview forum: "Shapley Is Not All You Need: Sobol's Total Indices for Feature Selection and Performance Loss Estimation"
_ICLR.cc/2025/Conference — ICLR 2025 Conference Withdrawn Submission_

### Official Review · Reviewer_zHjh · 2024-10-29

**Soundness:** 1
**Presentation:** 2
**Contribution:** 1
**Rating:** 3
**Confidence:** 3

**Summary:**

This paper compares two classical measures, the Shapley values and the Sobol’s total indices, in the context of feature selection.  It shows the superiority of the Sobol's total indices over the Shapley values for feature selection.

**Strengths:**

The experimental comparisons showed that the Sobol’s total indices might be better than the Shapley values for feature selection.

**Weaknesses:**

The motivation of this paper is unclear: the feature selection has important applications when handling high-dimensional data, but we do not use the Shapley value for such high-dimensional data due to its time complexity.  I do not understand why this paper chooses the Shapley values for feature selection despite the fact that there are so many faster alternative methods as discussed in Section 5.  The motivation for handling low-dimensional data is not described.  If the data is low-dimensional, we can directly compute Equation (1) instead of estimating it with the Sobol's total indices.

It should also be noted that this paper is not the first paper to compare the Shapley values and the Sobol's total indices, and this paper should discuss what are already known and what are new insights found in this paper.  Examples of existing work include:
+ B. Iooss and C. Prieur, Shapley effects for sensitivity analysis with correlated inputs: comparisons with Sobol' indices, numerical estimation and applications, International Journal for Uncertainty Quantification, 2019.
+ B. Vuillod et al., A comparison between Sobol's indices and Shapley's effect for global sensitivity analysis of systems with independent input variables, Reliability Engineering & System Safety, 2023.

The proof for Inequality (9) is missing.  Moreover, this inequality does not hold, because we have $\Delta(x_i)=0$, $\phi_i=0.25$, and $S_{T_i}=0$ according to the numbers in Table 3 for the Synthetic Correlated Dataset.

The discussion on the Shapley values on the synthetic datasets (lines 351-387) is inappropriate.  The Shapley values are not designed to estimate the performance loss, and therefore the obtained values 0.25 are perfectly fine.

The experimental results on the real datasets are not convincing to support the claim that the Sobol's indices are better.  I do not see significant difference between the prediction performances in Fig. 3, taking into account the confidence intervals.  In addition, Tables 5-7 do not contain any information on the confidence intervals.

**Questions:**

N/A

---

### Official Review · Reviewer_cM5K · 2024-11-01

**Soundness:** 1
**Presentation:** 2
**Contribution:** 2
**Rating:** 3
**Confidence:** 5

**Summary:**

In this paper, the authors claims the drawbacks of the Shapley value and claims the superiority of Sobol's total index. Specifically, the authors pointed out that the Shapley value suffers from high computational complexity and lacks an appropriate way to account for the impact of feature removal. Additionally, they noted that some Shapley value implementations are not model-agnostic. The authors argued that Sobol's total index is a preferable alternative that does not have these limitations.

**Strengths:**

A strength of this study is that it demonstrates the potential effectiveness of Sobol's total index as a feature attribution method. Sobol's total index is a simple approach, and, as the authors claim, it is highly advantageous in terms of computational complexity. In situations where Sobol's total index is effective, this advantage is likely to be significant.

**Weaknesses:**

The weakness of this study is that it only contrasts the Shapley value with Sobol's total index despite many feature attribution methods have already been proposed. Furthermore, as various studies indicate, feature attribution methods do not estimate a single, unified concept of "feature importance"; rather, each method defines and measures "feature importance" differently. Because these definitions vary, the usefulness of each method depends heavily on its targeted application. A method may be unsuitable for one purpose yet useful for another.

This study, however, does not account for the existing diverse literature of feature attribution methods. Instead, it narrowly compares the Shapley value and Sobol's total index from the authors' specific perspective of utility. While the Shapley value is indeed a popular method and identifying its limitations is valuable, this does not necessarily mean that Sobol's total index is inherently useful. To support the utility of Sobol's total index, it is essential to compare it with other major feature attribution methods from multiple perspectives. The evaluation in the current study remains highly limited.

**Questions:**

What are the advantages of Sobol's total index over other major feature attribution methods besides the Shapley value?

---

### Official Review · Reviewer_saRr · 2024-11-02

**Soundness:** 2
**Presentation:** 2
**Contribution:** 3
**Rating:** 5
**Confidence:** 4

**Summary:**

This paper tries to address feature selection in interpretable machine learning, with a focus on high-dimensional data. It advocates for Sobol’s total indices, arguing that they more accurately capture the variance reduction associated with feature removal while being model-agnostic and computationally efficient, scaling linearly with the number of dimensions.

**Strengths:**

1. The paper raises a good point regarding variable selection, noting that Sobol’s indices offer more accurate assessment of variable reduction compared to Shapley values.
2. The authors offer clear and intuitive explanations of the phenomena discussed.
3. The running time is an advantage compared to Shapley values, especially in high dimension.

**Weaknesses:**

1. The table oversimplifies the limitations of Shapley values-based methods. Model-dependent approaches like TreeSHAP do not suffer from factorial growth in time complexity, and methods such as SAGE [1] are model-agnostic. The table should present a more balanced view of the strengths and weaknesses of different Shapley-based methods rather than exclusively highlighting their disadvantages.
2. It’s better to explicitly mention how the Shapley values are calculated as there are many methods available. Current popular methods like SAGE are significantly more efficient than the method used in empirical experiments,  which required 9111.088 seconds.
3. It’s better to highlight Table. 5-7 or include plots like Fig. 3, 4 for a more straightforward comparison. Notably, in Table 7, Shapley values begin to outperform Sobol’s indices overall when feature number reaches 9, suggesting Shapley value may have a better selection ability in this data.
4. The related work section in Section 5 is somewhat confusing, as it deviates from the central discussion. A more cohesive organization could improve the article’s flow.

Editing issues:
1. broken line at line 378, page 8
2. broken line at line 432, page 9
3. Fig4. touches the left border

[1]. Covert, Ian, Scott M. Lundberg, and Su-In Lee. "Understanding global feature contributions with additive importance measures." Advances in Neural Information Processing Systems 33 (2020): 17212-17223.

**Questions:**

1. Eq (8) doesn’t seem to be well defined given the definition based on the preceding paragraph, and I couldn’t find a proof for Eq (9), could the authors clarify?
2. Eq (10) doesn’t seem to hold based on the definition of Eq (3), are the authors trying to use $S_{rT_3}$ defined in line 264?

---

### Official Review · Reviewer_vLGY · 2024-11-05

**Soundness:** 2
**Presentation:** 3
**Contribution:** 1
**Rating:** 3
**Confidence:** 4

**Summary:**

This paper studies Sobol' total indices as an alternative to Shapley values for feature selection in high-dimensional data. Through arguments and empirical studies that highlight that Sobol’ provides a more accurate measure of feature importance, particularly with highly correlated features or large interaction effects, via inclusion of their variance contributions. It is mentioned that interactive variances cannot fully be captured for with Shapley values, that tend to average over all features providing a more balanced view of how variance is shared amongst features. The authors conduct empirical experiments using both simulated and real datasets to determine that Sobol’ indices offer better computational efficiency and improved performance in feature selection as opposed to Shapley values, which are largely used today.

**Strengths:**

First, the paper is overall well-written and straightforward to follow, with clear explanations and a bolded logical structure that enables easier reading. Second, the authors provide a number of different experiments in comparing Sobol’ indices to Shapley values across a range of datasets, from simulations of known correlation structures (aligning with their argument that capturing interactive variance losses via Sobol’ indices is stronger than that of Shapley), to studies on runtimes. These studies thus allow the reader to observe the performance of each method in different contexts, which to the best of my knowledge has not been empirically studied in much detail before. Last, the visualizations, including those of runtime comparisons and accuracy trends, are clear and effective in illustrating these comparisons.

**Weaknesses:**

There are a number of flaws in the presented methodology, those which may stem from my own confusion regarding certain definitions and paragraphs.
My main critique is that I am struggling to see the novelty of results in this paper beyond the empirical studies which largely reflect what has been theoretically proved in the literature over the past decade or so. Specifically, it is well known that shapley values are derived from sobol indices such that Sobol’ indices bound shapley values, which can indeed be seen by definition and variance inclusion. Both metrics measure different but related aspects of features, but this has been well studied in the literature, and the idea of using Sobol’ indices is not new by construction.
Second, there appears to be some confusion in newly defining accuracy and other ML terminologies as they relate to feature importance. For example, from the sentence beginning “The actual performance loss for regression models and classification models are scaled differently”: Regression models typically output continuous values, whereas classification models classify discrete classes with assigned probabilities, thus it is to be expected that performance losses are scaled differently with respect to the underlying probability measure (which may include the empirical to align with model generality) encoded into the loss. Even between two different regression models, the performance losses can be different. I am thus a little confused as to why this paragraph is included in the paper, not what it is trying to convey to the reader, but perhaps I am missing something here. The claim that these models require different scaling adjustments is to me, not sufficiently justified, and thus this section would benefit from a clearer rationale for why scaling adjustments are necessary in this study.
Third, the authors argue against the need for the efficiency axiom in Sobol’ indices, where Shapley values conform to efficiency (and other nice theoretical properties including symmetry and additivity) to not just capture enough of the variance rather to provide a fair attribution of feature contributions across all subsets of features. The authors argue that efficiency is not needed when computing feature importance due to the focus on evaluating how much a feature contributes to overall performance as opposed to dividing predictive power equitably across features. However, I believe this stems from the goal of the practioner and state of the training data, as opposed to the philosophical definition of feature importance. An example here is whether to use \ell_1 or \ell_2 penalizations on a loss function to predict feature weights. For highly correlated features, one may prefer the \ell_2 to balance all features with similar weights reflecting their correlations, but when there are more features than data points, one may prefer to use the \ell_1 to zero out some correlated features to ensure computational robustness, effectively providing importance via sparsity. I therefore disagree with the authors’ views on not needing efficiency, as I believe it is highly problem dependent and the choice of the practitioner (who may not have this choice, in for example, the case where there are more features than data points).
Forth, the simulated correlated and XOR datasets, while powerful, only really test basic interactions. Such experiments would be stronger with additional datasets that may capture more complex or higher-order interactions, that align with the authors’ claims about Sobol indices’ interaction-handling capabilities.
Last, the authors claim that the efficiency of computing the Sobol’ indices is much higher than that of Shapley and provide nice empirical comparisons. However, this section does not include comparisons with the authors’ aforementioned Shapley approximations (such as Kernel SHAP) which is normally used to handle high dimensionality in real datasets. Further, the claim that the authors make on line 210 that such methods including Kernel SHAP and Tree SHAP are often model dependent is incorrect and misleading. Kernel SHAP is nonparametric and model agnostic.

**Questions:**

1) Please can you describe the paragraph beginning with “The actual performance loss for regression models and classification models are scaled differently” in more detail,  and what its utility is in the broad scope of the paper?
2) Please can you give a more balanced overview of Shapley vs Sobol in the sense of either theoretical and/or emprical properties hosted by one and not the other, or both at the same time?

---

### Official Review · Reviewer_JcmM · 2024-11-06

**Soundness:** 3
**Presentation:** 4
**Contribution:** 2
**Rating:** 5
**Confidence:** 3

**Summary:**

This paper proposes using Sobol's total indices instead of Shapley values for feature selection in machine learning models for three reasons:

Sobol's total indices capture both main effects and interactions, offering a more accurate importance measure than Shapley values.

Sobol's total indices are more computationally efficient as the computation scales linearly with the number of features.

Sobol's total indices are fully derived from the data itself.

This paper compares Shapley values and Sobol’s total indices for feature selection in both synthetic and real-world datasets to showcase Sobol’s total indices' advantages on feature selection.

**Strengths:**

1. The diagram (fig1), together with the argument in sections 1 and 2, gives readers a clear sense/insight into how these two metrics work and how they differ.

2. The code is lightly annotated and easy to understand.

3. The comparison between the two metrics is thorough ranging from all aspects, like definition, computation, interpretation, practical usage, etc.

**Weaknesses:**

1. One minor suggestion on notation: for variables with bar or hat on the top, maybe it is better to use \widehat or \widebar accordingly based on the width of the variable. For example: accuracy in eq 7 and \delta(x_i) in line 157-158.

2. Eq 8 is a bit misleading. I believe you are suggesting the difference in terms of R^2 and the difference in terms of accuracy right? Maybe writing \delta = R^2 or \delta = accuracy is not the best way.

3. Based on your code (function sobol_total_indices in method.py), it seems that the way you calculate (3) is by grouping y based on each unique value of x for each feature of interest. Should you treat continuous and categorical variables the same? if feature x is continuous, likely, group_y will only have length 1 or some very small size for each unique x value. Wouldn't this be an issue?

4. It would be much clearer if the authors could provide a detailed procedure (maybe an algorithm) on how \
    (1) experiments are conducted.\
    (2) how synthetic datasets are generated.

5. I don't think this is a fair comparison between Shapley and Sobol.  Shapley considers the average effect of one feature under all possible model selections while Sobol only measures the difference between the full model and the leave-one-feature-out model. They are not essentially taking care of the same thing. I am not sure about this part. Please correct me if I am wrong.

6. My last question is about the experiment design in general. From my perspective, there is a bit of disconnection between your claim and the experiments you do.

(1) For Table 6. I believe the concrete dataset is well-known for nonlinear relationships. Derived features like water-to-cement ratio and water-to-aggregates ratio are more important than those raw features. Also, since the paper emphasizes performance on high-dimensional datasets, wouldn't it be more reasonable to generate more interaction features first and then do the same analysis?

(2) similar to my previous point, I don't think any of the realistic datasets you include are high-dimensional. I believe it would be way more convincing if there could be high-d examples.

(3) Quote from Section 6:

"While Sobol’s total indices work better for feature selection tasks, they might not interpret a machine-learning model as well as the Shapley values do. Therefore, based on our findings, we conclude that Sobol’s total indices are better suited for feature selection in machine learning applications."

I agree with the interpretation part. But the second half might need more justification. Since you are already doing feature selection and more specifically backward feature selection on linear regression and decision trees, would it be better to include AIC, BIC (for LR), or Mean Decrease in Impurity (for trees) into the competition and see if Sobol can still outperform these classic metrics?

**Questions:**

Please see the Weaknesses.

---

### Note · Authors · 2024-11-24

**Comment:**

We appreciate the reviewers' effort and time spent providing feedback. We have decided to withdraw this work from this venue.

**Withdrawal Confirmation:**

I have read and agree with the venue's withdrawal policy on behalf of myself and my co-authors.